# Bisphenol A Analogues in Food and Their Hormonal and Obesogenic Effects: A Review

**DOI:** 10.3390/nu11092136

**Published:** 2019-09-06

**Authors:** Natalia Andújar, Yolanda Gálvez-Ontiveros, Alberto Zafra-Gómez, Lourdes Rodrigo, María Jesús Álvarez-Cubero, Margarita Aguilera, Celia Monteagudo, Ana Rivas

**Affiliations:** 1Department of Nutrition and Food Science, University of Granada, Campus of Cartuja, 18071 Granada, Spain; 2Department of Analytical Chemistry, University of Granada, Campus of Fuentenueva, 18071 Granada, Spain; 3Department of Legal Medicine and Toxicology, University of Granada, 18071 Granada, Spain; 4Department of Biochemistry & Molecular Biology III, University of Granada, PTS, 18016 Granada, Spain; 5GENYO, Pfizer-University of Granada-Junta de Andalucía Centre for Genomics and Oncological Research, Av. de la Ilustración 114, 18016 Granada, Spain; 6Department of Microbiology, Faculty of Pharmacy, University of Granada, Campus of Cartuja, Granada 18071, Spain; 7Instituto de Investigación Biosanitaria ibs, 18016, Granada, Spain

**Keywords:** bisphenol A analogues, food, obesogenic effect

## Abstract

Bisphenol A (BPA) is the most well-known compound from the bisphenol family. As BPA has recently come under pressure, it is being replaced by compounds very similar in structure, but data on the occurrence of these BPA analogues in food and human matrices are limited. The main objective of this work was to investigate human exposure to BPA and analogues and the associated health effects. We performed a literature review of the available research made in humans, in in vivo and in vitro tests. The findings support the idea that exposure to BPA analogues may have an impact on human health, especially in terms of obesity and other adverse health effects in children.

## 1. Introduction

Endocrine disruptors are compounds that alter the normal functioning of the endocrine system, and their bioaccumulation in humans may cause adverse health effects [1,2,3]. Bisphenol A (BPA) is a well-known endocrine disruptor, industrially produced, largely used as a component of epoxy resins and polycarbonate plastics [4,5]. BPA-based plastics and resins are used in the manufacturing of food contact material such as packaging, crockery, and thermic paper. Human exposure to BPA occurs mainly through diet (food and food contact materials) [6]. Moreover, BPA has also been found in plastic food containers, epoxy coatings in metal cans, kitchenware toys, medical devices, and dental composites and sealants [4,7,8,9,10].

In humans, BPA has proven to have developmental, reproductive, cardiovascular, immune, and metabolic effects [11]. In 2017, BPA was listed in the substances of very high concern list of the European Chemical Agency (ECHA). In view of the recent regulations that further restrict the use of BPA in food contact materials [12,13,14,15,16], food packaging companies are exploring substitutes to gradually eliminate BPA from their products [10,17,18].

Commercialization of BPA-free labeled products is increasing, while BPA analogues are being increasingly used in the manufacturing of consumer products [10]. BPA analogues share the basic bisphenol structure of two benzene rings separated by a short carbon or other chemical chain [10] (Figure 1). Bisphenol S (BPS), bisphenol F (BPF), bisphenol B (BPB), bisphenol E (BPE), and bisphenol AF (BPAF) are chosen by the industry as a replacement for BPA in the production of polycarbonates and epoxy resins [10,19,20] for the manufacturing of industrial and consumer products [21,22]. There are a limited number of studies on the BPA analogues’ hormonal effects, but most show that they have similar health concerns as BPA [23,24,25,26]. In 2002, we demonstrated the endocrine-disrupting activity of BPA analogues in the expression of estrogen-controlled genes [27]. Furthermore, some of the BPA substitutes seem to have more estrogenic effects than BPA [28]. Despite the regulatory actions taken in recent years, one potential hazardous chemical is being replaced with others with similar activity and health outcomes.

The health effects of BPA analogues have been reported recently in different studies [28,30]. Pelch et al., (2019) [25] concluded in a recent review that BPA analogues have health or toxicological effects at concentrations similar to or lower than BPA. BPA analogues have proven to have bone [31], reproductive [32], metabolic [33], oxidative stress [34], and neurological effects [35].

The considerable use of BPA analogues and their potential health risks require studies to better understand the complex and widespread routes of human exposure. The objective of this work was to identify data on the presence of BPA analogues in food, human biomonitoring data, and health effects of these chemicals, especially those related to obesogenic outcomes. This was addressed by a literature review of the available research. We hope the information in this review may be useful to move to new regulations based on classes of chemicals rather than on an individual chemical approach.

## 2. Methods

A review of the available literature was conducted in March 2019. PubMed/Medline and Scopus databases were searched using the keywords “Bisphenol A analogues”, “Bisphenol A substitutes”, “food”, “hormone effect”, “biological samples”, “health”, and “obesogenic effect”. Data published between 1999 and 2019 were considered.

Figure 2 shows the Preferred Reporting Items for Systematic Reviews and Meta-Analyses (PRISMA) flow diagram of the literature search [36]. Study selection consisted of two screening phases. The first selection was based on title and abstract screening, and the second selection was based on a full-text screening. One reviewer (N.A.) conducted the two screening phases, and in case of doubt, a second reviewer (A.R.) was consulted. The literature search yielded 572 studies. Studies were selected for full-text screening when they met the inclusion criteria. In case of doubt, articles were also analyzed based on their full text. Cross references were checked for additional references. Studies were excluded after the analysis of full text if they met one of the following criteria: (a) not English language; (b) not outcome of interest, and (c) not about the subject of the study.

## 3. Bisphenol A Analogues in Food

There is little information regarding the occurrence of bisphenol analogues (BPs) in foodstuffs. Liao and Kannan [37], in a study performed in the United States, determined the presence of BPA, BPF, and BPS (*N* = 267) in nine categories of foodstuffs and found that 75% of the samples contained BPs, with total concentrations ranging between below the limit of quantification (LOQ) and 1130 ng/g fresh weight (4.38 ng/g overall mean value). The highest overall mean concentration (sum of eight BPs) was detected in preserved and ready-to-eat foods. The highest BPF and bisphenol P (BPP) concentrations were 1130 ng/g and 237 ng/g, respectively, and were found in a sample of mustard and ginger. In contrast, BPs in beverages and fruits were found in concentrations of 0.341 ng/g and 0.698 ng/g, respectively. Higher levels of individual and total BPs were detected in canned food than in foods that came in plastic, glass, or paper containers.

Liao and Kannan [38], in a study performed in China, determined the presence of eight BPs (*N* = 289) in 13 categories of foodstuffs using high-performance liquid chromatography-tandem mass spectrometry (HPLC-MS/MS). The most frequently found BPs were BPA and BPF, which were detected at mean value concentrations of 4.94 ng/g and 2.50 ng/g fresh weight, respectively. The highest overall concentration (sum of eight BPs) was found in canned products (27.0 ng/g), followed by fish and seafood (16.5 ng/g), and beverages (15.6 ng/g). In contrast, the lowest overall concentration was found in milk and dairy products, cooking oils, and eggs (2–3 ng/g). Higher total concentration levels were detected in canned foodstuffs (56.9 ng/g) than in foods coming in glass (0.43 ng/g), paper (11.9 ng/g), or plastic (6.40 ng/g) containers.

Other studies have reported on the presence of BPA analogues in canned vegetables, fruits, and soft drinks [29,39] as well as in honey [40,41], fish [41], and mustard [42]. BPF has been found in mustard from white mustard seeds in mg kg^−1^ levels and is a natural reaction product formed during its preparation. Mustard is one of the most widely used condiments worldwide and, according to some authors, it is the main source of BPF in humans, in Europe, and probably worldwide [42]. In addition to BPA, bisphenol A diglycidyl ether (BADGE) and bisphenol F diglycidyl ether (BFDGE) have been detected in different milk samples from supermarkets and dairy farms [43], and BPF and BPS have been detected in dairy products, meat and meat products, vegetables, and cereals [37].

Although BPA is the most studied BP, BADGE and BFDGE have also been detected in drinking water as these BPs are commonly used in the epoxy coatings applied in drinking water distribution systems. BPs from the coatings may be exposed to chemical oxidants (disinfectants) that have the potential to form by-products with enhanced or reduced estrogenic activity than the parent compounds [44]. Exposure to free chlorine results in rapid BP degradation (half-lives of BPs are between <1 min to 35 min) under typical conditions [44].

There is an increasing amount of research linking long-term, low-level exposure to BPA in early life and adverse health effects in infants and fetuses [22]. Breast milk is the main source of energy for babies under six months and therefore it may be used as a proxy of the internal exposure levels in mothers and fetuses. Niu et al. (2017) [22] found BPA, BPF, BPS, and BPAF in breast milk samples, with BPA being the most abundant BP, followed by BPF. Recently, our research group has reported on the presence of BPB, BPS, BPE, and BPP in baby food samples [10].

Table 1 summarizes the BPs found in different food samples analyzed, their concentrations, and the analytical techniques used for detection.

## 4. Bisphenol A Analogues in Biological Samples and Their Hormonal Effects

Data on BPA analogue occurrence in human samples are scarce. Human uridine 5’-diphosphate-glucuronosyltransferases are able to rapidly metabolize BPA analogues into the corresponding BPA-glucuronide (UGTs) [63,64,65]. BPA-glucuronides are rapidly excreted in urine in rats and humans [66] and determination of urinary levels is considered a good biomarker of exposure to BPs [67].

Liao et al., (2012) [23] determined the total concentration of BPS in 315 urine samples, collected from the U.S. and different Asian countries, and detected BPS in 81% of the samples analyzed with variations between countries. BPS concentrations were between below the LOQ (0.02 ng/mL) and 21 ng/mL (geometric mean (GM): 0.168 ng/mL). The highest GM concentration of BPS was detected in samples from Japan (1.18 ng/mL).

Ye et al., (2015) [68] determined the concentration of BPA, BPS, BPF, and BPAF in 616 archived urine samples collected anonymously from volunteers during eight years. BPA was the most frequently detected chemical (74%–99%), whereas BPAF was detected in fewer than 3% of the samples. The detection frequencies were in the ranges of 74%–99% for BPA, 42%–88% for BPF, and 19%–74% for BPS. GM concentrations were in the ranges of 0.36–2.07 μg/L for BPA, 0.15–0.54 μg/L for BPF, and <0.1 to 0.25 μg/L for BPS. Although BPF concentrations were generally lower than other BPs, the 95th percentile concentration of BPF was similar or higher than that of BPA in most of the samples. The authors conclude that the significant changes in GM concentrations of BPA and BPS are consistent with a decline in BPA exposure and an increase in BPS exposure.

Lehmler et al., (2018) [69] investigated the association between the presence of BPA, BPF, and BPS in urine samples from adults (*N* = 1808) and children (*N* = 868) and different demographic and lifestyle variables; the BPA, BPS, and BPF levels were 95.7%, 89.4%, and 66.5%, respectively. In adults, the median levels of BPA were higher (1.24 μg/L) than BPF and BPS levels (0.35 and 0.37 μg/L, respectively). In children, the median BPA levels were also higher (1.25 μg/L) than BPF and BPS levels (0.32 and 0.29 μg/L, respectively). The study revealed associations between the different BPs and gender, race, income, physical activity, smoking, and alcohol intake.

Exposure of pregnant women to BPs is a particular concern, as these chemicals pass from mother to infant via breast milk, making this matrix a main target for exposure assessment of critical subpopulations. In this framework, Deceuninck et al. (2015) [70] investigated the presence of a large group of BPA analogues in breast milk samples (*N* = 30) and detected BPS in one sample at a 0.23 μg/kg concentration, whereas the rest of the BPA analogues investigated were not detected. In a study by Niu et al. (2017) [22] also on human breast milk from Chinese mothers, BPF, BPS, and BPAF were found, where BPA was still the most frequently compound detected, followed by BPF. This is the first study reporting on the presence of BPF and BPAF in human breast milk.

Available studies on BPF and BPS quantification in other biological matrices are still limited. The detection frequency of these compounds in human serum is relatively low [71]. BPS has been found in maternal and cord blood serum [33]. More recently, Song et al. (2019) [72] found a detection frequency of more than 65% for BPA, BPAF, and BPF in serum samples collected from residents living near recycling facilities where e-waste was being dismantled, with GM concentrations of 3.2, 0.0074, and 0.062 μg/mL, respectively.

The presence of BPA analogues in human biological samples suggests that it may have an effect on the body. The estrogen activity is the most widely studied effect of BPA, and its effects on other hormonal receptors have also been reported [65]. However, limited studies have reported that BPA analogues have endocrine-disrupting activities similar to those of BPA [17]. The estrogenicity of BPs was first reported in 1998 using an E-SCREEN assay in cultures of the human breast cancer cell line MCF7 [73]. Later on, in 2002, we demonstrated the effects of these chemicals on the expression of estrogen-controlled genes by measuring the induction of pS2 (mRNA and protein) and progesterone receptor as well as the expression of a luciferase reporter gene transfected into MVLN cells [27]. Subsequent studies have corroborated these findings [21]. It has also been reported that the estrogenic effect of some BPs is higher than that of BPA [74]. For example, BPS shows higher hormonal activity, which can likely be attributed to its strong polarity and the presence of a sulfonyl group [39,75] and its heat stability and resistance to light [10,70]. Moreover, BPS and BPF have been shown to be involved in breast cancer progression as much as BPA by inducing proliferation and migration of MCF-7 clonal cells [76]. Recently, Van Leeuwen et al., (2019) [77] reported that most BPA analogues in in vitro studies have similar or higher estrogenic activity than BPA as well as higher antiandrogenic properties. Other BPA analogues showed both antiestrogenic and antiandrogenic activity.

BPS and BPF are the most studied bisphenol analogues. A systematic review [28] that included 32 studies (25 in vitro and 7 in vivo) revealed that the potency of BPF and BPS was in the same order of magnitude as BPA and had similar hormonal effects. Additionally, the review showed that BPS and BPF had hormonal effects beyond those of BPA, such as changes in organ weights and enzyme expression levels. The authors concluded that BPS and BPF seemed to have similar potency and mechanisms of action to those of BPA, posing similar health effects. Other authors have also reported on the similarity between BPS and BPF and BPA in terms of their toxicological profiles, including metabolic, carcinogenic, and reproductive effects, as well as oxidative stress and DNA damage [18,28,78,79,80].

Some studies in animal models have also suggested the adverse reproductive effects secondary to exposure to BPA analogues, such as reduced sperm and oocyte quality and conversion of cholesterol into biologically active steroid hormones (steroidogenesis). These adverse effects depend on the duration of the exposure and the species investigated [81]. In this context, the effect of long-term BPF exposure on the reproductive neuroendocrine system in zebrafish has been recently demonstrated [82]. Similarly, in the same animal model, BPS was shown to decrease gonad weight and alter plasma estrogen and testosterone, as well as to reduce egg production and hatchability, with longer hatch periods, and increase embryo malformations. Shi et al., (2018) [83] showed that prenatal exposure to BPA analogues with physiologically relevant doses affects male reproductive functions probably due to a spermatogenic defect in the developing testis. Ullah et al. (2019) [84] demonstrated the impact of low-dose chronic exposure to BPB, BPF, and BPS on hypothalamo–pituitary–testicular activities in adult rats. The effect on female reproductive functions in mice after prenatal exposure to BPA analogues was recently demonstrated [85]. The authors concluded that prenatal exposure to bisphenols accelerated the onset of puberty, and the mice exhibited fertility problems, abnormal estrous cyclicity, and dysregulated expression of steroidogenic enzymes, especially with lower doses. Kolla et al., (2018) [86] compared the BPA and BPS exposure effect during the perinatal period on female mouse mammary gland development. The study revealed age- and dose-specific effects of BPS that were different from the effects of BPA. In addition, Zhou (2018) [87] evaluated low-concentration BPS toxicity using L1 larvae of the model animal *Caenorhabditis elegans*. Multiple indicators at the physiological, biochemical, and molecular levels were tested. Compared with the effects of BPA, the overall results showed that BPS was less noxious, suggesting that individual bisphenols may have unique effects. Other studies have also demonstrated the estrogenic, androgenic, and thyroidogenic activities of BPF and BPS [28,71].

Eladak et al., (2015) [30] developed a fetal testis assay to demonstrate that 10 nmol/L BPS and BPF can reduce the basal testosterone secretion by fetal human and mouse testes. Furthermore, more recently, Desdoits-Lethimonier et al., (2017) [88] showed that, using an ex vivo culture system, BPE, BPF, BPB, and BADGE exhibited antiandrogenic properties in adult human testes. In addition, in a study conducted on GH3 rat cell line, BPA, BPAF, BPB, BPF, BPS, and bisphenol Z (BPZ) have been found to alter the activity of the thyroid endocrine system, which seems to be increased by 17β-estradiol [89].

Table 2 summarizes the studies reporting on the hormonal effects of BPs.

## 5. Obesogenic Effects of Bisphenol A Analogues

There is increasing evidence that a number of chemicals can interfere in hormonal metabolism and regulation of adipocyte function. This may result in imbalanced hormone levels, which can lead to obesity. These obesity-promoting compounds are known as “obesogens” [95].

It seems that genetic predisposition, unhealthy dietary habits as well as a sedentary lifestyle alone are not responsible for the global epidemic of overweightedness and obesity [95].

At a molecular level, another mechanism by which obesogens could lead to weight gain or obesity is through the activation of nuclear transcription factors, such as the peroxisome proliferator-activated receptors alpha, delta, and gamma (PPARα, PPAR-δ, and PPAR-γ) and steroid hormone receptors, that regulate adipocyte proliferation and differentiation and lipid metabolism, thereby influencing body composition. The transcription factors bind to response elements in the DNA-regulating specific patterns of gene expression [95]. Wassenaar et al., (2017) [96] conducted a systematic review of the literature and found a significant positive association between early-life exposure to BPA and fat weight and triglycerides.

BPA promotes adipogenesis, adipose tissue inflammation, and alteration of glucose and lipid metabolism [97]. BPA has the ability to bind to human and animal PPARγ, which could trigger obesogenic activities [98]. PPARγ is highly expressed in adipose tissue and regulates adipocyte development and the uptake of lipids by adipocytes. Obesogens can cause obesity through direct activation of PPARγ, but other mechanisms involve indirect activation of the receptor by increasing the PPARγ protein and making it available to promoters of genes in the adipogenic pathway [99]. Measurements of the effects of BPA exposure on obesity should consider not only the body mass index (BMI) but also the role of BPA in adipogenesis, lipid, and glucose dysregulation and the presence of adipose tissue inflammation, which in turn results from the increased secretion of proinflammatory compounds [97].

As with the hormonal effects of BPA substitutes, their obesogenic activity has not been well studied. However, the ability of BPS to promote accumulation of lipids and differentiation of human preadipocytes through the activation of a PPARγ pathway has been described [100]. Damage in the transcriptome of preadipocytes during their differentiation has been described in relation to long-term exposure to low doses of BPA or BPS and BPF [101]. As with BPA, BPS could induce differentiation of preadipocytes. Both BPA and BPS are weak activators of PPARγ and need this receptor to induce adipogenesis [102]. BPA and BPS can increase the differentiation of 3T3-L1 preadipocytes in mice in a dose-dependent manner. Moreover, BPS has shown stronger adipogenic activity than BPA [102].

Ivry Del Moral et al., (2016) [103] showed that BPS can increase the harmful effects of a high-fat diet and induce changes in the postprandial lipid metabolism, increasing fat accumulation in the adipose tissue.

Recently, Charisiadis et al., (2018) [104] investigated the relation between obesity and the presence of BPA and BPF in hypothalamic and white matter postmortem material from 12 pairs of obese (BMI >30 kg/m^2^) and normal-weight individuals. A significant association (*p* < 0.05) between BPA and BPF concentration and obesity was found, except for BPF in white matter samples.

In another study by Liu et al. (2017) [33] conducted in 1521 adults (≥20 years) who participated in the National Health and Nutrition Examination Survey 2013–2014, significant associations were found between BPA exposure and general and abdominal obesity, whereas no significant associations were found for BPF or BPS, at the current exposure level. However, the authors suggested continued biomonitoring of these BPA substitutes.

In addition, a recent study by Liu et al., (2019) [62] showed for the first time that exposure to BPF was positively associated with a higher risk of obesity in children and adolescents. This association was mainly found in boys, suggesting a potential gender influence.

Table 3 summarizes the studies reporting on the obesogenic effect of BPs.

## 6. Conclusions

Given the similarities between BPA and BPA analogues in terms of their metabolism and actions, including hormonal effects beyond those of BPA, it is little surprise that these substitutes also represent a risk to wildlife and human health. Regulations on the assessment of the safety of consumer products should be extended to cover all the compounds of the same chemical group. In addition, as recommended by different researchers, more efforts are needed to find chemical substitutes without hazardous health effects. The substitution trend toward BPA analogues in consumer products, particularly in food contact materials, should be implemented with caution and include effective and regular monitoring to assess their effects on human health.

## Figures and Tables

**Figure 1 nutrients-11-02136-f001:**
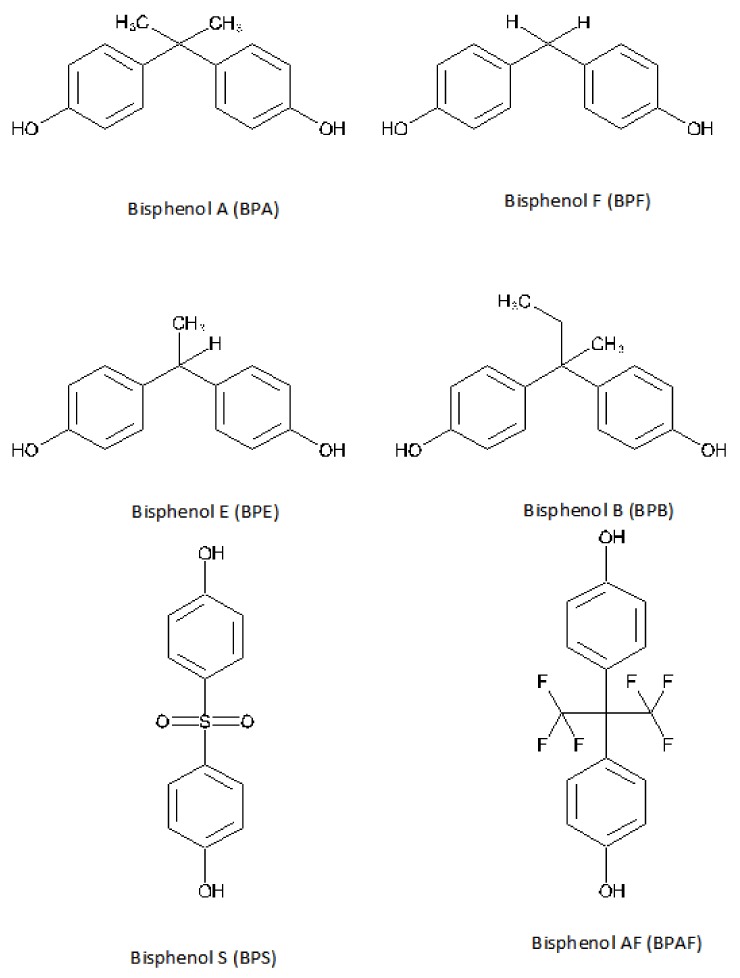
Bisphenols with endocrine-disrupting activity. Prepared by author based on Gallart-Ayala et al., 2011 [29].

**Figure 2 nutrients-11-02136-f002:**
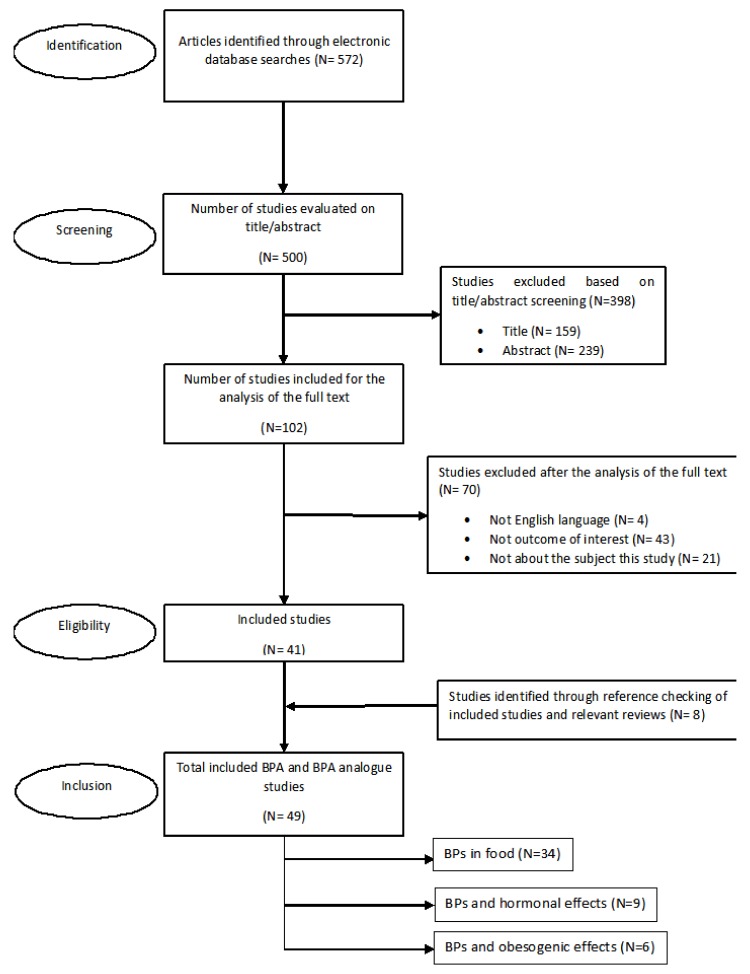
PRISMA flow diagram of the literature search.

**Table 1 nutrients-11-02136-t001:** Bisphenol analogues (BPs) found in food samples using different analytical techniques.

Reference	Food	Bisphenol	Concentration	Analytic Technique
Abou-Omar et al., 2017 [45]	27 olive oils	BPA	non-plastic packaging Mean = 333 μg/kg plastic packaging Mean = 150 μg/kg	High Performance Liquid Chromatography with Fluorescence Detection (HPLC-FLD)
Alabi et al., 2014 [46]	Canned vegetables (mushroom, red pepper, olive, green beans, asparagus), legumes (chickpeas, lentils), canned fruits (pineapple, peach), canned fish and other seafood (mackerel, mussels, tuna, cockles), canned meat products (tripe, meat ball), and canned grains (sweet corn)	BPA	13–241 μg/kg	HPLC-FLD
		BPF	ND	
		BADGE	7.1 μg/kg	
		BPE	ND	
		BPB	25–40 μg/kg	
		BFDGE	21–314 μg/kg	
Cacho et al., 2012 [5]	Canned beverages and filling liquids of canned vegetables	BPA	0.10–0.35 μg/L	Stir bar sorptive extraction combined with thermal desorption (Gas chromatography–mass spectrometry, GC-MS)
		BPF	0.4 μg/L	
		BPZ	ND	
Cesen et al., 2016 [40]	36 honey samples	BPA	Mean = 107 ng/g	GC-MS
		BPAF	Mean = 53.5 ng/g	
		BPE	Mean = 12.8 ng/g	
		BPF	Mean = 31.6 ng/g	
		BPS	Mean = 302 ng/g	
		BPZ	Mean = 28.4 ng/g	
Cirillo et al., 2015 [47]	Infant formulas	BPA	Median = 0.015 μg/g	HPLC-FLD
Cunha et al., 2011 [48]	30 beverages and 7 powdered infant formulas	BPA	Mean = 0.115 μg/L	Dispersive liquid–liquid microextraction and GC-MS
		BPB	Mean = 2.365 μg/L	
Cunha et al., 2012 [49]	47 canned seafood samples (22 tuna, 10 sardines, 3 mackerel, 3 squid, 3 octopus, 2 mussels, 1 eel, 1 anchovy, 1 cod)	BPA	1.0–99.9 μg/kg	QuEChERS (dispersive solid-phase cleanup) combined with dispersed liquid–liquid microextraction and GC-MS
		BPB	21.8 μg/kg (only one sample)	
Cunha and Fernandes, 2013 [50]	20 canned vegetables and 19 canned fruits	BPA	Mean = 265.6 μg/kg	QuEChERS combined with GC-MS
		BPB	Mean = 3.4 μg/kg	
Fasano et al., 2015 [51]	3 packed grated cheese, 2 meat, 2 fish, and 2 vegetables broths, 2 white and 2 red wines, 4 pasta, 4 rice, 3 chicken and vegetables	BPA	0.72–218 ng/g	QuEChERS combined with GC-MS
Feshin et al., 2012 [52]	Fruit and vegetable puree, canned fruit and vegetables; milk, meat puree, and canned meat	BPA	2.15–42.91 ng/g	GC-MS
Gallart-Ayala et al., 2011a [39]	11 canned soft drinks (soda, cola, tea, energy drink, and beer)	BPA	44–607 ng/L	Liquid chromatography-tandem mass spectrometry (LC-MS/MS)
		BPF	218–141 ng/L	
Gallart-Ayala et al., 2011b [29]	6 canned foods (vegetables and fruits) and 7 canned beverages (soda, cola, tea, and tonic drink)	BADGE	2.3–675 μg/kg	LC-MS/MS
		BFDGE	ND	
Gallo et al., 2017 [18]	40 canned energy drink samples	BPA	0.50–3.3 ng/ml	Ultra-high performance liquid chromatography linked with fluorescence detection (UPLC-FLD)
		BADGE	0.50–19.4 ng/ml	
		BFDGE	0.50–0.60 ng/ml	
García-Córcoles et al., 2018 [10]	15 baby food samples	BPS	11.7–49.2 ng/g	GC-MS/MS
		BPB	1.1–8.5 ng/g	
		BPP	1–7.7 ng/g	
Grumetto et al., 2013 [53]	68 milk samples were packed in Tetra Pack or Tetra Brix boxes or in plastic bottles (PET, PEHD)	BPA	5.21–14.0 ng/ml	LC-FLD
		BPF	0.1–26.2 ng/ml	
		BPB	16.0–67.0 ng/ml	
Kuo and Ding, 2004 [54]	4 soy-based infant formula powders	BPA	45–113 ng/g	GC-MS
Lapviboonsuk and Leepipatpiboon, 2014 [55]	3 canned tuna	BADGE	ND	QuEChERS and HPLC
Liao and Kannan, 2013 [37]	31 beverages, 29 dairy products, 5 fats and oils, 23 fish and seafood, 48 cereals and cereal products, 51 meat and meat products, 20 fruits and canned fruits, 45 vegetables and canned vegetables, and 15 “others”	BPA	0.285–9.97 ng/g	HPLC-MS/MS
		BPAF	0.005–0.021 ng/g	
		BPAP	0.005–0.185 ng/g	
		BPB	0.013–0.017 ng/g	
		BPF	0.025–4.63 ng/g	
		BPP	0.013–0.562 ng/g	
		BPS	0.005–0.609 ng/g	
		BPZ	0.025–0.076 ng/g	
Pardo et al., 2006 [56]	10 canned pig meat	BADGE	83–87 ng/g	Reversed-phase HPLC coupled to atmospheric pressure chemical ionization tandem mass spectrometry
		BFDGE	96–101 ng/g	
Rastkari et al., 2010 [57]	12 canned tomato paste and 12 canned corn	BPA	0.90–47.38 μg/kg	GC-MS
		BPF	0.89–47.11 μg/kg	
Rauter et al., 1999 [58]	142 canned oily foods	BADGE	0.02–1.5 mg/kg	HPLC combined with GC-MS
Sadeghi et al., 2015 [41]	Canned fruits (pineapple, peach), canned vegetables (tomato), powdered milk, soft drinks, honey, and fish	BPA	0.9–8.3 ng/g	HPLC-FLD
Simoneau et al., 2012 [59]	449 plastic baby bottles	BPA	0.5–1000 μg/kg	HPLC and UPLC-MS combined with LC-MS
Viñas et al., 2010 [60]	Canned foods (peas, peas with carrots, sweet corn, artichoke, mushroom, bean shoot, and mixed vegetables)	BPA	11.7–321 ng/mL supernatant	GC-MS in the selected ion monitoring
			12.9–77.7 ng/g food	
		BPS	8.90–175 ng/g supernatant	
			34.1–36.1 ng/g foods	
Xionga et al., 2018 [43]	Milk samples from supermarkets and dairy farms	BPA	14.31 µg/kg	QuEChERS and HPLC-FLD
		BADGE 2H2O	15.80 µg/kg	
		BFDGE 2H2O	16.23–17.82 µg/kg	
Yang et al., 2014 [61]	2 coconut juice samples	BPS	0.019–0.036 ng/g	LC-MS/MS
		BPA	0.23–12 ng/g	
		BPF	0.39 ng/g	
		BPAF	0.013–0.052 ng/g	
Zhang et al., 2010 [62]	Canned fish and meat	BADGE	58.76–140.72 ng/g	HPLC-FLD
		BFDGE	40.57–77.64 ng/g	
Zoller et al., 2016 [42]	61 mustard samples	BPF	Mean 1.85 mg/kg	LC-MS/MS, liquid chromatography/high resolution mass spectrometry (LC-HRMS) and gas chromatography/high resolution mass spectrometry GC-HRMS)
Zou et al., 2012 [45]	3 canned porridge, 3 canned mushroom, 3 canned *Cirrhinus molitorella*, 3 canned tuna, 3 canned anchovy, 3 canned pork, 3 canned pork sauce, 3 canned peanut butter	BADGE	1.78–88.08 ng/g	UPLC-MS/MS
		BFDGE	2.13–32.96 ng/g	

ND, not detected; HPLC-FLD, high-performance liquid chromatography, fluorescence detector; GC, gas chromatography; MS, mass spectrometry; UPLC, ultra-performance liquid chromatography; HRMS, high-resolution mass spectrometry.

**Table 2 nutrients-11-02136-t002:** Studies about hormonal effects of BPs.

Reference	Species Strain Mode	Dose Exposure	Exposure Route	Outcomes	Conclusion
Desdoits-Lethimonier et al., 2017 [88]	Human	1^−^⁹ to 10^−^⁵ M for 24 or 48 h (BPA, BPF, BFS, BFE, BPB, and BADGE)	In vitro adult testes from prostate cancer patients	Significant dose-dependent inhibition between testosterone levels in the culture medium and concentrations of BPA and analogues. BPA and analogues induced inhibition of testosterone production with variations based on duration of exposure and BPA/analogue concentrations. Germ cells were not affected by BPA and analogues.	Direct exposure to BPA or its analogues can result in endocrine alteration in adult human testes.
Eladak et al., 2015 [90]	Mouse, human	10, 100, 1000, and 10,000 μmol/L (BPA, BPF, and BPS) for 48 h	In vitro human fetal testes, in vitro mouse fetal testes	BPS or BPF reduced basal testosterone secretion. BPS or BPF decreased Insl3 expression.	BPS and BPF result in adverse effects on testes of mice and humans.
Kim et al., 2017 [76]	Human	10 ^−^⁹ to 10 ^−^⁵ mol/L (BPA, BPF, and BFS) for 24 h	In vitro human breast cancer cells	BPA, BPS, and BPF increased proliferation of MCF-7 CV cell line by regulating the protein expression of cell cycle-related genes and epithelial mesenchymal transition (EMT) markers via the ER-dependent pathway.	BPS and BPF are associated with the increased risk of breast cancer progression as much as BPA in the proliferation and migration of MCF-7 CV cells.
Mokra et al., 2018 [91]	Human	0.01, 0.1, and 1 μg/mL for 4 h (BPA, BPS, BPF, and BPAF) 0.001, 0.01, and 0.1 μg/mL for 48 h (BPA, BPS, BPF, and BPAF)	In vitro peripheral blood mononuclear cells (PBMCs)	After 48 h, damage was present and change in PBMCs (peripheral blood mononuclear cell) viability exposed to BPA, BPS, BPF, and BPAF for 48 h (from 0.001 to 0.1 μg/mL for which a decrease in cell viability did not exceed 20%).	BPA, BPS, BPF, but mostly BPAF, caused oxidative damage to DNA in pyrimidine bases and more strongly to purine bases in human PBMCs. Confirmation of BPA and BPA analogues being strongly genotoxic.
Molina-Molina et al., 2013 [92]	Human	0.01 to 10 μM (BPA, BPS, and BPF) for 4 h	In vitro cells from leukocyte-buffy coat	BPS, BPF, and BPA activated estrogen receptors. BPS was more active in the estrogen receptor beta. BPF and BPA were full androgen receptor agonist.	BPA and its analogues affect non-genomic signaling in estrogen-responsive cells, with potential consequences for cell function.
Naderi et al., 2014 [93]	Zebrafish	0, 0.1, 1, 10, and 100 μg/L for 75 days (BPS)	In vivo embryos	Gonadosomatic index was reduced (≥10 μg/L) and hepatosomatic index increased. Plasma 17β-estradiol levels (≥1 μg/L) were increased and testosterone showed a reduction in males (10 and 100 μg/L). An induction in plasma vitolegenin level was observed (≥10 μg/L). Egg production and sperm count were decreased (10 and 100 μg/L). Postponed and decreased rates of hatching were observed.	Exposure to low doses of BPS has adverse effects on different parts of the endocrine system in zebrafish.
Roelofs et al., 2015 [79]	Mouse	10 to 300 μM (BPA) for 48 h 0.01 to 100 μM (BPF) for 48 h 10 μM (BPS) for 48 h 10 to 100 μM (TBBPA) for 48 h	In vitro Leydig cell line M^−10^	BPA and BPF presented glucocorticoid receptor (GR) and androgen receptor (AR) antagonism with IC₅₀ values of 67 μM, 60 μM, and 22 nM for GR, and 39 μM, 20 μM, and 982 nM for AR, respectively, whereas BPS did not affect receptor activity. Testicular steroidogenesis was altered by all BPs tested. BPF and BPS increased the levels of progestogens that are formed in the beginning of the steroidogenic pathway.	BPF and BPS induce Leydig cell testosterone secretion and GR antagonism in the nanomolar range.
Rosenmai et al., 2014 [78]	Human	0.3 to 100 μM (BPA, BPB, BPE, BPF, BPS, and BPP) for 24 h	In vitro adrenal cortico-carcinoma cells	BPS presented less estrogenic and antiandrogenic activity than BPA. BPS showed the largest efficacy on17α-hydroxyprogesterone.	BPA analogues interfere with the endocrine system.
Stroheker et al., 2003 [94]	Rat	0, 25, 50, 100, and 200 mg/kg bw/day (BPA and BPF) for 21 days	In vivo system	BPA did not induce an increase in relative wet or dry uterine weight. BPF induced a significant dose-related increase in relative wet uterine weight at 100 mg/kg bw/day and above and a significant increase of relative dry uterine weight at 200mg/kg bw/day.	Exposure to BPF has weak estrogenic effects in rats. In immature rats, the effects are more sensitive, inducing uterine growth.

**Table 3 nutrients-11-02136-t003:** Studies about obesogenic effects of BPs.

Reference	Species Strain Model	Dose Exposure Period	Exposure Route	Outcomes	Conclusion
Ahmed and Atlas, 2016 [102]	Mouse	0.01–50 μM (BPS) and 0.01–50 μM (BPA) for 2 days	In vitro 3T3-L1 cells	BPS promotes the expression of adipogenic markers and lipid storage. Treatment of 3T3-L1 cells with BPS can increase lipoprotein lipase, adipocyte protein 2, PPARγ (Peroxisome proliferator-activated receptor), perilipin, adipsin, and enhancer-binding protein alpha mRNA expression levels. BPS and BPA can weakly activate PPARγ using a PPARγ. BPS but not BPA was able to competitively inhibit rosiglitazone activated PPARγ.	BPS is a more potent adipogen than BPA. BPA and BPS can upgrade 3T3-L1 adipocyte differentiation in a dose-dependent manner and require PPARγ to adipogenesis.
Boucher et al., 2016 [100]	Human	0.1 nM to 25 μM (BPS) for 14 days	In vitro primary preadipocytes	25 μM BPS induced lipid accumulation, increased the mRNA and protein levels of several adipogenic markers, including lipoprotein lipase and adipocyte protein2 (aP2). BPS did not affect lipoprotein lipase protein levels.	BPS promotes lipid storage and differentiation of primary human preadipocyte.
Ivry Del Moral et al., 2016 [103]	Mouse	0.2, 1.5, 50 μg/kg bw/day (BPS) for 23 weeks	In vivo system	BPs induced overweightedness in male mice offspring fed with a high-fat diet at the two highest doses of BPs. Obesity was related to hyperinsulinemia, hyperleptinemia, and fat mass. Plasma triglycerides were significantly increased with BPs. Finally, BPS induced alteration in mRNA expression of marker genes involved in adipose tissue homeostasis.	BPS potentiates obesity in high-fat diets by inducing lipid storage linked to faster lipid plasma clearance.
Liu et al., 2017 [33]	Human	Median urinary concentrations were: −0.7 to 1.47 ng/L BPA −0.1 to 1.3 ng/L BPF −0.2 to 1.0 ng/L BPS	Human urine	Associations between BPA levels and obesity were stronger in men (OR = 2.10; 95% CI: 1.20–3.68) than in women and for white (OR = 2.41; 95% CI: 1.32–4.43) than non-white participants. The associations of BPS and BPF with general obesity did not differ by sex or ethnicity.	There was association of BPA exposure with general obesity and abdominal obesity.
Riu et al., 2011 [98]	Human, zebrafish, *Xenopus*	10^−9^ to 10^−4^ M (BPA, TetrabromoBPA and tetrachloroBPA)	In vitro estrogen receptors (ERs) and PPARs	TetrabromoBPA and tetrachloroBPA are human, zebrafish, and *Xenopus* PPARγ ligands and determine the mechanism by which these chemicals bind to an activated PPARγ. Activation of ERα, ERβ, and PPARγ depends on the degree of halogenation in BPA analogues. Bulkier brominated BPA analogues have a greater capability to activate PPARγ and a weaker estrogenic potential.	Polyhalogenated BPs could function as obesogens acting as agonists to disrupt physiological functions regulated by human or animal PPARγ.
Verbanck et al., 2017 [101]	Human	10 nM and 10 μM (BPA, BPS, and BPF) for 10 days	In vitro primary preadipocytes from subcutaneous fats	Chronic exposure of preadipocytes to BPA, or its substitutes BPS and BPF, results in deleterious effects on their transcriptome during differentiation of human primary adipocytes, even at a low dose (10 nM).	Caution required over the use of BPA, BPS, and BPF, since unsuspected cell damage could be initiated at low doses.

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
