# Peer review of "Bisphenol A Analogues in Food and Their Hormonal and Obesogenic Effects: A Review"

_nutrients, 2019, doi:10.3390/nu11092136_

Round 1
Reviewer 1 Report
It is a very good comprehensive review about BSA and its effects. Nevertheless, several concerns should be added. Introductoin section shouold be more informative and elaborate about the BSAs and the effects that could be done in health. In which categorization belong the BSA? Updated references should be added, for example Int J Environ Res Public Health. 2018 May 30;15(6) and Toxics. 2017 Dec 21;6(1). pii: E2. The section with the detection of BSA should be merged with the section of the effects. I think that there are more effects of BSA on female/male reproduction.
Author Response
August 15, 2019
Editor in chief
Dear
REVIEWER 1
It is a very good comprehensive review about BSA and its effects. Nevertheless, several concerns should be added.
1.Introduction section should be more informative and elaborate about the BSAs and the effects that could be done in health.
Author´s response:
Following the reviewer´s suggestion, we have improved the Introduction section including more and recent information about Bisphenol A analogues health effects.
2.In which categorization belong the BSA?
Author´s response:
The authors thank the reviewer for this interesting observation. We understand that the reviewer suggested that Bisphenol A analogues are categorized as endocrine disrupters chemicals. Following the reviewer suggestion we have explained that in the Introduction section of revised manuscript.
On the other hand, only bisphenol A was classify as a category 1B presumed reproductive toxicant, by the European Commission in 2016 meaning it is a substance which can adversely affect the human reproductive system.
In 2017, the Member State Committee of European Chemical Agency identify Bisphenol A as a substance of very high concern because of its endocrine disrupting properties which cause probable serious effects to human health. This is explained in the Introduction section of our manuscript.
In addition, a new working group of scientific experts from the European Food Safety Authority (EFSA) will start evaluating recent toxicological data on bisphenol A in food contact material. Details on this new assessment are due in 2020.
In addition, The International Agency for Research on Cancer (IARC) Classified two bisphenol A derivatives as group 2A (tetrabromobisphenol) and group 3 (Bisphenol A diglycidyl ether).
3.Updated references should be added, for example, Int J Environ Res Public Health. 2018 May 30;15(6) and Toxics. 2017 Dec 21;6(1). pii: E2.
Author´s response:
As recommended, we have included update references (including Int J Environ Res Public Health. 2018 May 30;15(6) and Toxics. 2017 Dec 21;6(1). pii: E2) in the Introduction section of revised manuscript.
4.The section with the detection of BSA should be merged with the section of the effects.
Author´s response:
As suggested, the sections with the detection and effects of BPA analogues have been merged in the revised Manuscript.
5.I think that there are more effects of BSA on female/male reproduction.
Author´s response:
As recommended, we have included more studies on BPA analogues effects on female/male reproduction.

Reviewer 2 Report
Overall this paper is interesting and does an adequate job in reviewing the literature on BPA and its analogues. More research needs to be done on BPA analogues, and thus this review may be useful to readers. However, some concerns are listed below.
Major concerns:
There are several studies showing their similar estrogenic effects that you have not cited. You keep saying limited studies show that the analogues have toxic effects, and this implies that there are papers showing the analogues are safe. It would be better to say that there are limited papers on the analogous but most show that they have similar health concerns as BPA. This might be related to English wordings.
Another concern is how you determined whether a paper was related or not. Your criteria sounded subjective.
The text boxes in figure 2 are partially cut-off; Table 1 headings should realign better. Some places, the font is bigger than others (L150, L178 etc).
Minor comments/edits:
Line 25-28 (grammar)
We performed a literature review of the available research made in humans, in vivo and in vitro tests. The findings support the idea that exposure to BPA analogues may have an impact on human health, especially in terms of obesity and other adverse health effects in children
Line 38-39 (grammar)
In humans, BPA has proven to have developmental, reproductive, cardiovascular, immune and metabolic effects.
Line 43 (grammar)
Commercialization of BPA-free labeled products is increasing.
Line 47
Is BPAF really being used as an alternative given that it is more toxic than BPA?
Line 158-159 (grammar)
Sentence is wordy and redundant, consider revising
sources to consider:
SriDurgaDevi Kolla, Mary Morcos, Brian Martin, Laura N. Vandenberg,
Low dose bisphenol S or ethinyl estradiol exposures during the perinatal period alter female mouse mammary gland development,
Reproductive Toxicology,Volume 78,2018, Pages 50-59, ISSN 0890-6238,
https://doi.org/10.1016/j.reprotox.2018.03.003.
Dong Zhou. Ecotoxicity of Bisphenol S to Canerhabditis elegans by Prolonged Exposure in Comparison with Bisphenol A. Enviromental Toxicology. 2018, 37,10.

Author Response
REVIEWER 2
Overall this paper is interesting and does an adequate job in reviewing the literatura on BPA and its analogues. More research needs to be done on BPA analogues, and thus this review may be useful to readers. However, some concerns are listed below.
Major concerns:
There are several studies showing their similar estrogenic effects that you have not cited. You keep saying limited studies show that the analogues have toxic effects, and this implies that there are papers showing the analogues are safe. It would be better to say that there are limited papers on the analogous but most show that they have similar health concerns as BPA. This might be related to English wordings.
Author´s response:
The authors thank the reviewer for this interesting observation. We agree with the reviewer that this part of the Introduction is confusing. It is due to English wordings. There are different and recent studies that show hormonal activities of BPA analogues. In our revised manuscript, we have included more studies confirming the endocrine disrupting activity of BPA analogues. In addition, we have revised the Introduction section clarifying and enhancing the readability of our text.
Another concern is how you determined whether a paper was related or not. Your criteria sounded subjective.
Author´s response:
We agree with the reviewer that the selection of studies was not well explained. We have now included further information in the revised Method section.
The text boxes in figure 2 are partially cut-off;Author´s response:
Following the reviewer´s suggestion, the text boxes in figure 2 have been amended.
Table 1 headings should realign better.Author´s response:
Following the reviewer´s suggestion, the Table 1 headings have been realign better.
Some places, the font is bigger than others (L150, L178 etc).
Author´s response:
The Font was corrected in the revised manuscript.
Minor comments/edits:
6.Line 25-28 (grammar)
We performed a literature review of the available research made in humans, in vivo and in vitro tests. The findings support the idea that exposure to BPA analogues may have an impact on human health, especially in terms of obesity and other adverse health effects in children
Author´s response:
All of the grammar improvements recommended by the reviewer have been made.
7.Line 38-39 (grammar)
In humans, BPA has proven to have developmental, reproductive, cardiovascular, immune and metabolic effects.
Author´s response:
The sentence has been corrected in the revised manuscript.
3.Line 43 (grammar)
Commercialization of BPA-free labeled products is increasing.
Author´s response:
The sentence has been corrected in the revised manuscript.
4.Line 47
Is BPAF really being used as an alternative given that it is more toxic than BPA?
Author´s response:
The authors thank the reviewer for this interesting observation. Different methods, including in vitro, in vivo, and molecular docking have showed that the hormonal activity of BPAF is stronger than that of BPA (Lei et al., 2017. Environ. Toxicol. 32: 278e289; Mesnage et al., 2017. Toxicol. Sci. 158: 431e443; Mu et al., 2018. Environ. Sci. Technol. 52: 3222e3231).
However, BPAF has widespread application as a curing agent in the production of epoxy resins, in monomer in polycarbonate copolymers, electronic materials, and so on (Song, et al., 2012. Environ. Sci. Technol. 46:13136e13143; Karrer et al., 2019. Environ Sci Technol. 53:9181-9191.). BPAF is now also recognized as one of the replacements for BPA in the production polycarbonate plastic and resins because the concerns regarding the toxicological effects of BPA (Lei et al., 2019. Chemosphere 220: 362-370; Karrer et al., 2019. Environ Sci Technol. 53:9181-9191.)
5.Line 158-159 (grammar)
Sentence is wordy and redundant, consider revising
Author´s response:
Following the reviewer´s suggestion, the sentence has been revised
Sources to consider:SriDurgaDevi Kolla, Mary Morcos, Brian Martin, Laura N. Vandenberg,
Low dose bisphenol S or ethinyl estradiol exposures during the perinatal period alter female mouse mammary gland development,
Reproductive Toxicology,Volume 78,2018, Pages 50-59, ISSN 0890-6238,
https://doi.org/10.1016/j.reprotox.2018.03.003.
Dong Zhou. Ecotoxicity of Bisphenol S to Canerhabditis elegans by Prolonged Exposure in Comparison with Bisphenol A. Enviromental Toxicology. 2018, 37,10.
Author´s response:
Following the reviewer´s suggestion, these sources have been included.
